# Effects of Drought and Rehydration on the Physiological Responses of *Artemisia halodendron*

**Juanli Chen** [1,2,*], **Xueyong Zhao** [1], **Yaqiu Zhang** [3], **Yuqiang Li** [1], **Yongqing Luo** [1] , **Zhiying Ning** [1,2], **Ruixiong Wang** [1,2], **Peiyu Wang** [4] **and Anqi Cong** [1,2]

[1]  Naiman Desertification Research Station, Northwest institute of Eco-Environment and Resources, Chinese Academy of Sciences, Lanzhou 730000, China; zhaoxy@lzb.ac.cn (X.Z.); liyq@lzb.ac.cn (Y.L.); luoyongqing8401@sina.com (Y.L.); ningzhiying125@163.com (Z.N.); wangrx2015@lzu.edu.cn (R.W.); angelcong1990@163.com (A.C.)

[2]  Northwest Institute of Eco-Environment and Resource, University of Chinese Academy of Sciences, Beijing 100049, China

[3]  Horticultural Technology Department, Hanzhong Agricultural Technology Extension Center, Hanzhong 723000, China; zhangyaqiu1003@163.com

[4]  College of Geography and Environment, Northwest Normal University, Lanzhou 730070, China; 18356620807@163.com

\*  Correspondence: juanlic@163.com; Tel.: +86-153-9311-4308

**Abstract:** *Artemisia halodendron* is a widely distributed native plant in China's Horqin sandy land, but few studies have examined its physiological responses to drought and rehydration. To provide more information, we investigated the effects of drought and rehydration on the chlorophyll fluorescence parameters and physiological responses of *A. halodendron* to reveal the mechanisms responsible for *A. halodendron*'s tolerance of drought stress and the resulting ability to tolerate drought. We found that *A. halodendron* had strong drought resistance. Its chlorophyll content first increased and then decreased with prolonged drought. Variable chlorophyll fluorescence (*F*v) and quantum efficiency of photosystem II (*F*v/*F*m) decreased, and the membrane permeability and malondialdehyde increased. When plants were subjected to drought stress, superoxide dismutase (SOD) activity degraded under severe drought, but the activities of peroxidase (POD) and catalase (CAT) and the contents of soluble proteins, soluble sugars, and free proline increased. Severe drought caused wilting of *A. halodendron* leaves and the leaves failed to recover even after rehydration. After rehydration, the chlorophyll content, membrane permeability, SOD and CAT activities, and the contents of the three osmoregulatory substances under moderate drought began to recover. However, *F*v, *F*v/*F*m, malondialdehyde, and POD activity did not recover under severe drought. These results illustrated that drought tolerance of *A. halodendron* resulted from increased enzyme (POD and CAT) activities and accumulation of osmoregulatory substances.

**Keywords:** Horqin sandy land; hydration-dehydration; chlorophyll fluorescence; lipid peroxidation; antioxidant enzyme activity; solute accumulation

## 1. Introduction

About one-third of the world's area is arid or semi-arid, but nearly half of China's terrestrial area belongs to arid or semi-arid regions [1]. Drought seriously affects plant growth and development, crop yield, and gene expression because of the importance of water in biochemical process [2–4]. As a result of global climate change, drought is becoming a more frequent and increasingly severe problem [5,6]. Since native plants are frequently subjected to water stress in drought-prone regions,

they have evolved many strategies and mechanisms to resist the effects of water deficits through long-term natural selection [7]. Consequently, plants in arid and semi-arid areas have developed a series of mechanisms for drought tolerance.

One group of responses relates to the fact that plants under stress generate large amounts of reactive oxygen species (ROS) that cause membrane lipid peroxidation, increased membrane permeability, and protein inactivation, among other changes, and these changes can lead to cell and plant death [8]. Plants respond to this stress by activating a protective system that scavenges ROS by increasing the activities of anti-oxidant enzymes such as peroxidase (POD), superoxide dismutase (SOD), and catalase (CAT) [9]. In addition, they synthesize large quantities of osmotic regulators to maintain cell turgidity by reducing rates of water loss [10,11].

Plants in different families and genera show different or even opposite responses to drought stress. For example, the chlorophyll content in the leaves of *Periploca sepium* [12] and *Hordeum vulgare* [13] increased with drought intensity, whereas the chlorophyll content [14], variable chlorophyll fluorescence (*Fv*) and quantum efficiency of photosystem II (*Fv/Fm*) decreased [15]. Under natural conditions, *Artemisia ordosica* accumulated osmotic regulators and increased the activity of its antioxidant enzymes during drought stress [16]. The malondialdehyde content and antioxidant enzyme activity of *Setaria viridis* and *P. sepium* increased with increasing drought intensity and decreased with irrigation during repeated drought and rehydration treatments [12,17]. Many studies have shown that the ability of psammophytes to withstand their harsh environment is closely related to their ability to strongly resist oxidation and to regulate cell osmotic contents.

Due to severe desertification over the past several decades, nearly 10% of the area in China's Horqin Sandy land is covered by mobile dunes. The common vegetation succession pattern in this region has three main periods: (1) the pioneer species *Agriophyllum squarrosum* becomes established on mobile dunes; (2) the dominant species *A. halodendron* colonizes the resulting semi-fix dunes; and (3) climax species such as *S. viridis* and *Leymus chinensis* become established on the resulting fixed dunes [18]. The successful establishment of *A. squarrosum* promotes the subsequent establishment of other species. When the dunes become semi-fixed, colonization by *A. halodendron* promotes a rapid increase in plant diversity and biomass [19]. *Agriophyllum squarrosum* and *A. halodendron* are therefore important species for vegetation restoration in sandy land, and there has been much research on *A. squarrosum*. However, *A. halodendron* has been studied much less often, even though this semi-shrub, which belongs to the Compositae, is widely distributed in three of northern China's major sandy lands: the Horqin, Hulunbuir and Songnen in the eastern part of Inner Mongolia [20]. In recent years, *A. halodendron* has received growing attention to understand its adaptation to the adverse conditions in these regions. Its responses to drought have been studied from the perspectives of ecological hydrology, population ecology, and community succession [19,21,22]. However, few studies have been reported on the physiological responses of *A. halodendron* to drought and rehydration [23]. To provide some of the missing data, we designed a study to examine the physiological adaptations of *A. halodendron* to drought to clarify how *A. halodendron* responds to water stress, and the relationships between these responses and the intensity of the stress. We hypothesized that *A. halodendron* would exhibit changes in its photosynthesis (chlorophyll fluorescence parameters), antioxidant system, and osmoregulation in response to increasing drought stress.

## 2. Materials and Methods

### 2.1. Experimental Site

We conducted our study at the Naiman Desertification Research Station (42°58′N, 120°43′E; 360 m asl), Chinese Academy of Sciences, which is located in the southeastern part of the Horqin Sandy Land, in eastern Inner Mongolia, China. This area has a continental semi-arid monsoonal climate and the mean temperature is 6.5 °C. The mean annual precipitation is 351.7 mm, with 70 to 80% of the total falling in the growing season. The landscape in this region is characterized by dunes and inter-dune

lowlands. Most of the lowland has been reclaimed for agriculture, and the vegetated dunes are used as pastures. The sandy soil mainly consists of coarse sand and silt. The natural vegetation mainly consists of *A. squarrosum*, *A. halodendron*, *S. viridis*, *Corispermum marocarpum*, *Salsola collina*, *Caragana microphylla*, *Bassia dasyphylla*, and *Tribulus terrestris* [24].

## 2.2. Experimental Design and Treatment Description

We collected 2-year-old *Artemisia halodendron* Turcz. ex Besser seedlings, with a mean height of 50 cm, from semi-fixed dunes and transplanted them into plastic pots (21 cm in height and 28 cm in diameter) containing dune soil on 25 May 2018. The saturated water content of the sandy soil was 21.1% and the field capacity was 13.0%. We planted 240 plants in 120 pots and cultivated them until late July to let them recover from any transplanting damage, with well watering. Superfluous water drained through holes in the pot bottom.

On August 1, we randomly divided the pots into three groups for three different water treatments (i.e., n = 40 per treatment), with water provided at levels defined as a proportion of field capacity: (1) Control, in which plants were watered to maintain them at 60 to 65% of field capacity; (2) Moderate drought (MD), in which plants were watered to 40 to 45% of field capacity; (3) Severe drought (SD), in which plants were watered to 20 to 25% of field capacity. Soil water content was controlled within the prescribed range by weighing the pots daily for 10 days (from August 1 to August 10). Then, 60 pots (i.e., n = 20 per treatment) were selected randomly to determine the effects of the continuous drought stress on the physiological parameters of *A. halodendron* on August 11, 17, and 23. The remaining 60 pots (i.e., n = 20 per treatment) were watered to restore the soil water content to field capacity on August 17 and 23 for rehydration to test the impacts of rehydration after drought. The experiment was carried out in the field, under natural conditions, but using a rain shelter to exclude rainfall. The rain shelter reduced light intensity to about 90% of ambient levels, but was installed at a sufficient height (4 m) that air circulation was not greatly affected and temperatures were within 1 °C of the ambient temperatures outside the shelter.

## 2.3. Analytical Methods and Statistical Analysis

The surviving mature leaves (completely developed leaves from the middle part of the stem) were chosen to detect the physiological indices. On the sampling dates, we measured the chlorophyll fluorescence characteristics and then sampled the plants to determine antioxidant enzyme activities and osmoregulatory contents. The leaves of the first 60 pots were removed using scissors on August 11, 17, 23, and the leaves of the remaining 60 plants were sampled on August 18 and 24. Some of the plants were brought to the laboratory to immediately measure the leaf chlorophyll content and membrane permeability; the other samples were placed in liquid nitrogen for determining physiological indicators.

The leaf chlorophyll content was measured spectrophotometrically using a spectrophotometer (Shimadzu Corporation, Japan) after macerating the leaves in acetone [25]. Chlorophyll fluorescence parameters were determined by a chlorophyll fluorescence instrument (Hansatech, England) [26]. The initial fluorescence ($Fo$) was determined after dark adaptation of leaves for 30 min, and then the maximum fluorescence ($Fm$) was determined by irradiation of saturated pulses (2800 µmol/(m$^2$·s)). Subtracting $Fo$ from $Fm$ was used to calculate $Fv$.

Relative water content (RWC) of the leaves was measured with the following equation:

$$\text{RWC} = [(\text{FW} - \text{DW})/(\text{TW} - \text{DW})] \times 100\%. \tag{1}$$

FW is the weight of fresh leaf, DW is the dry weight after oven-drying leaves at 70 °C for 24 h, and TW is the turgid weight after rehydration in a petri dish covered with nylon mesh for 12 h until constant weight [27].

Leaf samples were extracted with buffer (2% polyvinylpolypyrrolidone, 50 mM phosphate) using a mortar and pestle; the extract was centrifuged at 15,000 *g* for 20 min and stored at 4 °C for analysis of

antioxidant enzyme activities, malondialdehyde and osmotic regulators contents. The malondialdehyde content was determined using the thiobarbituric acid method; the supernatant absorbance was noted at 600, 532 and 450 nm [28]. POD activity was determined spectrophotometrically by absorption at 470 nm based on the oxidation of guaiacol [29]. SOD activity was measured spectrophotometrically at 560 nm using the nitroblue tetrazolium photoreduction method [30]. CAT activity was determined at 240 nm using the iodine–hydrogen peroxide method [31]. The activity of antioxidant enzymes was indicated with units (U) per gram of fresh weight [32].

The soluble protein content was measured spectrophotometrically at 595 nm using the coomassie blue dye combination method [33]. The soluble sugars content was determined with the method described by An et al. [12]. The free proline concentration was measured spectrophotometrically at 520 nm using the ninhydrin method [34]. The soluble sugars and proline were expressed as μg per gram of FW, and the soluble proteins were expressed as milligram per gram of FW [17].

Statistical analysis of the data was conducted using version 20.0 of the SPSS software for Windows (https://www.ibm.com/analytics/spss-statistics-software). We used one-way analysis of variance (ANOVA) to test for significant differences among the treatments. When the ANOVA results was significant, we used the Scheffe (C) and S-N-K (S) tests to identify significant differences between pairs of treatments. The image was constructed using Sigma Plot 12.5 software. We used Pearson's correlation coefficient (r) to test for significance of the relationships among physiological characteristics.

## 3. Results

Because the plants were dead under the severe drought treatment on August 23 and 24, it was not possible to measure the related indicators.

### 3.1. Changes in Chlorophyll Fluorescence

Severe drought caused *A. halodendron* leaves to wilt on August 23, and they failed to recover even when they were re-watered. The chlorophyll content first increased slightly from August 11 to 17, and then decreased significantly from August 17 to 23 (Figure 1A). On August 18, rehydration caused a remarkable increase in the chlorophyll contents and the contents became higher than those before rehydration, and the difference was distinct in the control and under moderate drought; the chlorophyll content after rehydration was highest in the control (3.996 mg g$^{-1}$ FW, versus 3.685 and 3.486 mg g$^{-1}$ FW, respectively, in the moderate drought and severe drought). The chlorophyll content under moderate drought was markedly lower than the control on August 23 and 24. On August 24, leaf chlorophyll content was lower than that before rehydration, but the difference was not notable.

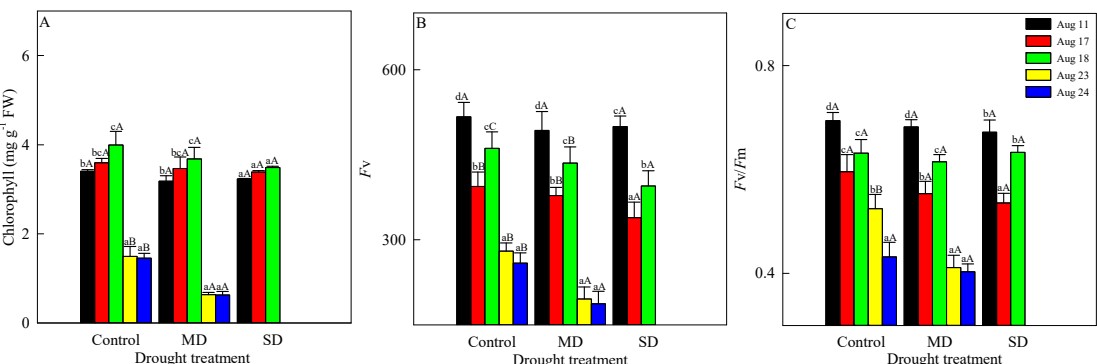

**Figure 1.** Changes in (**A**) leaf chlorophyll, (**B**) variable chlorophyll fluorescence (*F*v), and (**C**) quantum efficiency of photosystem II (*F*v/*F*m) of *A. halodendron* under drought and rehydration phase. Values represent mean ± SE. Bars labelled with different lowercase letters represent significant differences among days for a given treatment. Bars labelled with different capital letters differ significantly among drought treatments on a given day. 11, 17 and 23 August 2018 correspond to drought, 18 and 24 August 2018 correspond to rehydration.

*F*v decreased as drought duration and intensity increased (Figure 1B). The highest *F*v was found in the control on August 11, and the lowest was found under moderate drought on August 23. There was no significant difference between treatments on August 11, but by August 17, *F*v was markedly lower under severe drought than in the control and under moderate drought. On August 18, *F*v was dramatically higher than it was before rehydration, and *F*v differed markedly among the treatments. On August 24, *F*v decreased slightly but there was no significant difference from the values without rehydration, and was dramatically lower under moderate drought than in the control.

Drought stress decreased *F*v/*F*m (Figure 1C). The differences in *F*v/*F*m between the control and the other treatments were not remarkable until August 23, when *F*v/*F*m under moderate drought (0.485) decreased dramatically and became markedly lower than in the control (0.411). On August 18, *F*v/*F*m increased compared with the value before rehydration, but the difference among the treatments was not significant. On August 24, *F*v/*F*m continued to decrease, but the difference between the control and moderate drought was not notable.

### 3.2. Changes in RWC, Membrane Permeability and Malondialdehyde Content

Leaf RWC decreased with increasing drought duration and intensity (Figure 2A). On August 11, leaf RWC under moderate and severe drought was lower than that of the control, by 9.6% and 15.7%, respectively, and the differences among treatments were remarkable. On August 17, leaf RWC under severe drought was the lowest (54.8%). On August 18, re-watering caused a marked increase, a full recovery of leaf RWC was obtained, all values were greater than 87% and significantly higher than those on August 11 and 17, but there was no notable difference among treatments. On August 24, leaf RWC increased remarkably and was markedly higher than those on August 23, but dramatically lower than those on August 18.

The membrane permeability increased markedly with increasing drought duration and intensity (Figure 2B). On August 17, the membrane permeability had increased by 64.7%, 103.2% and 128.9%, respectively, compared with the corresponding values on August 11, and the differences among treatments were notable. The membrane permeability was highest under moderate drought (68.0%) on August 23, and this value was significantly greater than in the control. On August 18, the membrane permeability was markedly lower than before rehydration, but there was no remarkable difference among treatments. On August 24, membrane permeability was dramatically lower than that before rehydration. The membrane permeability was 51.1% under moderate drought, which equaled 2.43 times the control value.

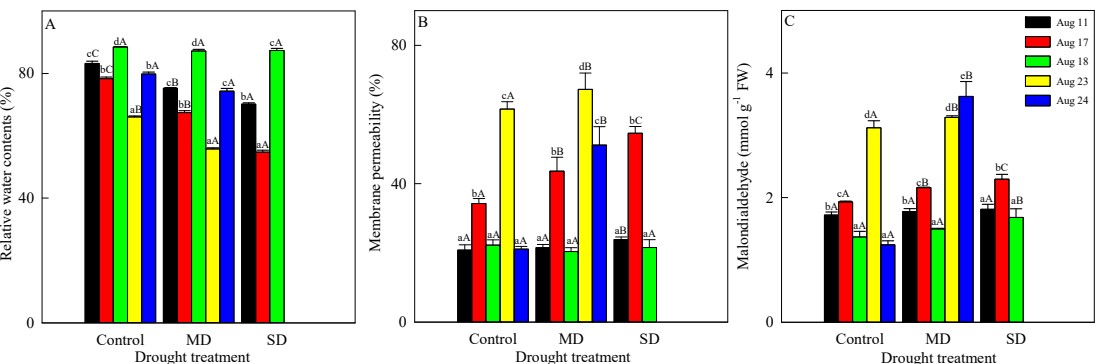

**Figure 2.** Changes in (**A**) Relative water contents (RWC), (**B**) membrane permeability, and (**C**) the malondialdehyde content of *A. halodendron* under drought and in the rehydration phase. Values represent mean ± SE. Bars labelled with different lowercase letters represent significant differences among days for a given treatment. Bars labelled with different capital letters differ significantly among drought treatments on a given day. 11, 17 and 23 August 2018 correspond to drought, 18 and 24 August 2018 correspond to rehydration.

Changes in the malondialdehyde content (Figure 2C) were consistent with the changes of membrane permeability, with values below 3.5 mmol·g$^{-1}$ FW in all treatments. The malondialdehyde content increased more from August 17 to 23 than it did from August 11 to 17, and both increases were remarkable. On August 17, malondialdehyde content levels under moderate and severe drought were both dramatically higher than in the control, but the content was markedly lower under moderate drought than under severe drought. On August 23, the malondialdehyde content in the control and moderate drought treatments increased by 52.2% and 61.7%, respectively, compared to the values on August 17. The difference between the control and moderate drought was remarkable. On August 24, the content in the control was significantly lower than the corresponding values before rehydration. The malondialdehyde content under moderate drought was 3.624 mmol·g$^{-1}$ FW, which was 2.91 times higher than the control and was significantly higher than before rehydration.

### 3.3. Changes in Antioxidant Enzymes Activity

POD activity was weak and the value was less than 0.5 U g$^{-1}$ FW (Figure 3A). From August 11 to 23, POD activity increased markedly with increasing drought duration and intensity. The POD activity under moderate drought on August 17 was significantly higher than in the control and under severe drought. POD activity under moderate drought on August 23 reached a maximum of 0.107 U g$^{-1}$ FW and was markedly greater than that in the control. After rehydration on August 18, POD activity decreased significantly in the two drought treatments, and the control values were markedly greater than those under moderate drought and severe drought. On August 24, POD activity in the control was dramatically lower than that before rehydration, but POD activity under moderate drought reached its maximum value of 0.133 U g$^{-1}$ FW, which was significantly greater than that in the control.

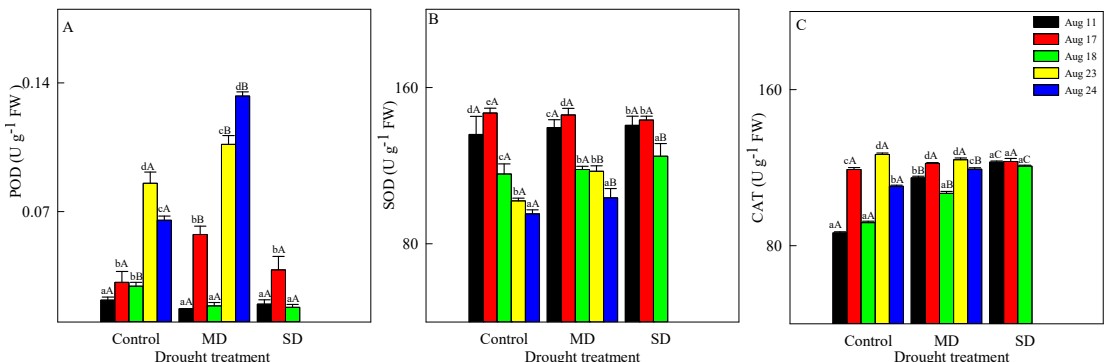

**Figure 3.** Changes in (**A**) peroxidase (POD), (**B**) superoxide dismutase (SOD), and (**C**) catalase (CAT) activity of *A. halodendron* under drought and in the rehydration phase. Values represent mean ± SE. Bars labelled with different lowercase letters represent significant differences among days for a given treatment. Bars labelled with different capital letters differ significantly among drought treatments on a given day. 11, 17 and 23 August 2018 correspond to drought, 18 and 24 August 2018 correspond to rehydration.

SOD activity increased markedly from August 11 to 17 in the control and under moderate drought, but there was no notable difference among the treatments (Figure 3B). From August 17 to 23, SOD activity decreased dramatically. On August 23, SOD activity was lowest in the control (101.956 U g$^{-1}$ FW), which was markedly lower than that (117.125 U g$^{-1}$ FW) under moderate drought. After rehydration on August 18, SOD activity decreased dramatically, and the activity was significantly greater under severe drought than under moderate drought and in the control. After rehydration on August 24, SOD activity decreased markedly under moderate drought and in the control.

CAT activity increased with increasing drought duration and intensity (Figure 3C). On August 17, CAT activity in the control and under moderate drought and severe drought increased by 37.6%, 6.5%, and 0.3%, respectively, compared with the values on August 11. On August 23, CAT activity was

highest in the control (126.825 U g$^{-1}$ FW). There was no remarkable difference among treatments on August 17 and 23. On August 18 and 24, CAT activity decreased dramatically compared to the values before rehydration, and CAT activity in the control was significantly lower than that under moderate and severe drought.

### 3.4. Changes in Osmoregulatory Substances

The content of soluble proteins increased significantly with increasing drought duration (Figure 4A). There was no remarkable difference among the treatments on August 17, but from August 17 to 23, the increase in the soluble protein content was much higher than that from August 11 to 17. On August 23, the soluble protein content was highest in the control and reached the highest value (38.746 mg g$^{-1}$ FW), which was markedly greater than that (37.323 mg g$^{-1}$ FW) under moderate drought and was 2.97 and 3.10 times, respectively, the value on August 11. On August 18, the soluble protein content was significantly lower than that before rehydration in all treatments, and the content under severe drought was evidently greater than those under moderate drought and in the control. The soluble protein content on August 24 was significantly lower than that before rehydration, but was significantly greater under moderate drought than in the control.

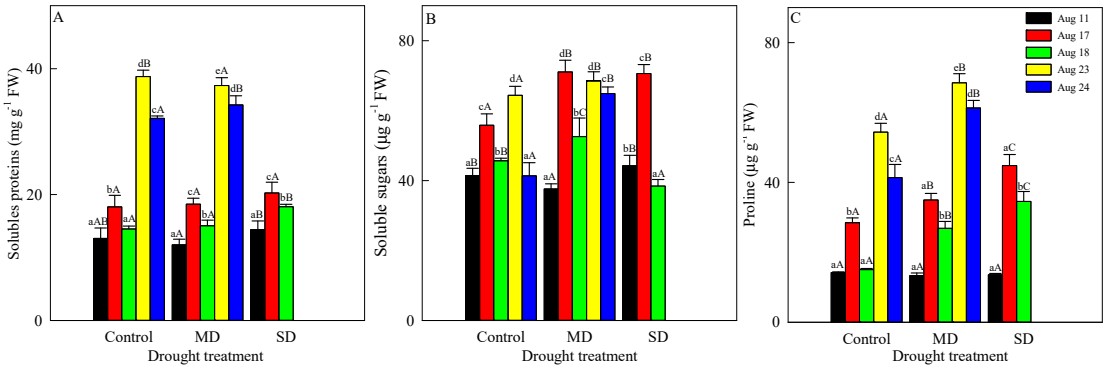

**Figure 4.** Changes in the (**A**) soluble proteins, (**B**) soluble sugars, and (**C**) free proline of *A. halodendron* under drought and in the rehydration phase. Values represent mean ± SE. Bars labelled with different lowercase letters represent significant differences among days for a given treatment. Bars labelled with different capital letters differ significantly among drought treatments on a given day. 11, 17 and 23 August 2018 correspond to drought, 18 and 24 August 2018 correspond to rehydration.

The soluble sugar content under moderate drought increased markedly from August 11 to 17, and then decreased slightly, but not significantly from August 17 to 23 (Figure 4B). The content in the control increased gradually with increasing duration of drought stress, but it was evidently lower than that in the other treatments on August 17 and 23. On August 18, the soluble sugar content decreased significantly compared with the values before rehydration, and there was a remarkable difference among drought treatments; the soluble sugar content in the control and under moderate drought was 45.684 μg g$^{-1}$ FW and 52.595 μg g$^{-1}$ FW, respectively, i.e., 18.9 and 36.8% higher, respectively, than that under severe drought. On August 24, the soluble sugar content was evidently less than that before rehydration.

During the drought period and after rehydration, the free proline content increased with increasing drought duration and intensity (Figure 4C). There was no significant difference among the treatments on August 11, but the difference was remarkable on August 17 and 23. The free proline content reached its maximum under moderate drought on August 23 (68.49 μg g$^{-1}$ FW), which was 5.11 times the value on August 11 and markedly greater than that in the control (25.9%). On August 18, the free proline content decreased significantly compared with the values before rehydration and differed evidently among the treatments. After rehydration on August 24, the free proline contents under moderate

drought and severe drought were 26.890 µg g$^{-1}$ FW and 34.551 µg g$^{-1}$ FW, respectively, and were higher than that in the control by 79.0 and 130.0%, respectively.

### 3.5. Correlation Analysis

We found significant correlations among the measured parameters (Table 1). *F*v was markedly positively correlated with *F*v/*F*m, but because the latter variable includes *F*v in its calculation, that correlation is not meaningful due to autocorrelation. However, both fluorescence parameters were evidently positively correlated with the chlorophyll content and SOD activity, but significantly negatively correlated with membrane permeability, malondialdehyde, POD activity, and the three osmotic regulators. RWC was markedly negatively correlated with membrane permeability, malondialdehyde, CAT activity, soluble sugars and proline. We also found remarkable positive correlations among POD activity, malondialdehyde, and membrane permeability, and all three of these parameters were markedly positively correlated with the three osmotic regulators, but significantly negatively correlated with the chlorophyll content and SOD activity. CAT activity was not markedly correlated with any of the other parameters. The chlorophyll content was significantly positively correlated with SOD activity, but significantly negatively correlated with the proline and soluble proteins contents. Proline was markedly positively correlated with membrane permeability, malondialdehyde, POD activity, soluble proteins, and soluble sugars.

**Table 1.** Pearson's correlation coefficient (*r*) for the relationships among the physiological characteristics for all drought durations including rehydration in *A. halodendron*. Abbreviations: $F_v$, variable chlorophyll fluorescence; $F_v/F_m$, quantum efficiency of photosystem II; RWC, relative water contents; MDA, malondialdehyde; POD, Peroxidase; SOD, superoxide dismutase; CAT, Catalase.

| Item | Chlorophyll | *Fv* | *Fv/Fm* | RWC | Membrane Permeability | MDA | POD | SOD | CAT | Soluble Proteins | Soluble Sugars |
|---|---|---|---|---|---|---|---|---|---|---|---|
| *Fv* | 0.868 ** | | | | | | | | | | |
| *Fv/Fm* | 0.865 ** | 0.984 ** | | | | | | | | | |
| RWC | 0.439 | 0.481 | 0.488 | | | | | | | | |
| Membrane permeability | −0.625 * | −0.741 ** | −0.714 ** | −0.831 ** | | | | | | | |
| MDA | −0.739 ** | −0.719 ** | −0.668 * | −0.643 * | 0.887 ** | | | | | | |
| POD | −0.911 ** | −0.909 ** | −0.912 ** | −0.471 | 0.758 ** | 0.851 ** | | | | | |
| SOD | 0.774 ** | 0.668 * | 0.594 * | −0.173 | −0.631 * | −0.844 ** | −0.789 ** | | | | |
| CAT | 0.032 | −0.066 | 0.020 | −0.663 * | 0.401 | 0.240 | −0.069 | −0.130 | | | |
| Soluble proteins | −0.924 ** | −0.932 ** | −0.925 ** | −0.474 | 0.739 ** | 0.730 ** | 0.895 ** | −0.731 ** | 0.197 | | |
| Soluble sugars | −0.369 | −0.614 * | −0.625 * | −0.735 ** | 0.884 ** | 0.725 ** | 0.627 * | −0.365 | 0.263 | 0.519 | |
| proline | −0.825 ** | −0.965 ** | −0.927 ** | −0.582 * | 0.846 ** | 0.807 ** | 0.879 ** | −0.732 ** | 0.227 | 0.911 ** | 0.709 ** |

Note: * and ** indicate significant difference at *P* < 0.05 and *P* < 0.01 levels, respectively.

## 4. Discussion

Chlorophyll is the main photosynthetic pigment and plays a crucial role in the absorption, transmission, and transformation of light energy. Its content affects the efficiency of carbon fixation by photosynthesis and plant drought resistance [35]. Under drought stress, the lamellar structure of chloroplasts can be damaged and the chlorophyll can be decomposed, thereby decreasing the chlorophyll content and causing yellowing of the leaves [14]. Within a certain range, the chlorophyll content directly affects the photosynthesis. We found that drought stress initially increased the chlorophyll content of *A. halodendron*, which differed from previous studies of less-drought-tolerant crops, which showed that drought stress decreased the chlorophyll content of spring wheat [36] and *Calotropis procera* [37], and this likely reflects *A. halodendron's* ability to resist drought stress by maintaining photosynthesis. Another study of cereal crops found that plants that could maintain higher chlorophyll content under drought stress could utilize light energy more effectively, thereby improving their drought resistance [38]. In the early stage of drought, we found the chlorophyll content increased slightly, which may indicate that *A. halodendron* could tolerate short-term drought, even if the drought was severe. However, the chlorophyll contents decreased significantly with increasing

drought duration and severity. Because plants produced a large number of ROS under drought stress, the cell membrane was destroyed and protoplast leaked, which accelerated chloroplast degradation and then inhibited chlorophyll synthesis [13].

Chlorophyll fluorescence provides important clues about the absorption, transmission, conversion, dissipation, and distribution of light energy during photosynthesis. It therefore helps to reveal the relationships between plant photosynthetic physiology and stress [39]. *F*v reflects reduction of the primary quinone electron acceptor (QA) in photosystem II (PSII) and represents the photochemical activity of PSII. In contrast, *F*v/*F*m reflects the primary light energy conversion efficiency, and its decrease indicates that the PSII reaction center has been damaged to some extent by stress [40]. *A. halodendron* showed similar trends in *F*v and *F*v/*F*m under drought stress: both values decreased markedly with prolonged drought stress, and the difference between treatments became significant for *F*v and *F*v/*F*m on August 23. This indicated that drought stress reduced the ability of QA and of the secondary quinone electron acceptor (−QB) to transmit electrons in the PSII reaction center, which led to inhibition of transmission by plastoquinone (PQ). At the same time, the photochemical activity of the PSII reaction center decreased, leading to an increase in the excess excitation energy [41], which would limit the normal progress of photosynthesis. Rehydration in the early stages of drought (on August 18), chlorophyll content and the values of *F*v and *F*v/*F*m increased compared to the corresponding values before rehydration. This indicated that rehydration alleviated drought stress and allowed *A. halodendron* to recover from the stress. After rehydration in the late stages of drought (on August 24), the leaf chlorophyll content did not differ markedly from the values before rehydration because the severity of the drought stress caused the leaves to wilt, and they failed to recover even after release of the stress by the added water.

Many studies have shown that the balance between production and removal of ROS in plant cells is destroyed by severe temperature, salt, radiation, or drought stress. The resulting accumulation of ROS causes peroxidation of membrane lipids, and malondialdehyde is the product of membrane lipid peroxidation [42]. This explains why the malondialdehyde content increased with increasing drought duration and severity. Malondialdehyde can react with macromolecules such as proteins and nucleic acids in cells, leading to denaturation of the proteins and nucleic acids and weakening of covalent bonds between cellulose molecules. Therefore, the malondialdehyde content represents the degree to which plant cells are damaged by stress [43].

The cell membrane is the main site where plant cells exchange substances with their external environment, and its stability directly affects the plant's metabolic functions. Therefore, the extent of changes in cell membrane permeability can be used as an important indicator of the cell's stress. Leaf RWC is also used to reflect the arid environment in which the plants are located and the degree of dehydration of the cells [15]. On August 11, there was no remarkable difference in the malondialdehyde content among the treatments, but the membrane permeability under severe drought was markedly higher than that in the control and under moderate drought. This indicated that even though the different intensities of drought stress did not lead to significant membrane lipid peroxidation in *A. halodendron* after water control, the cell membrane had already begun to show signs of damage under severe drought. With prolonged drought stress, leaf RWC decreased markedly, the membrane permeability changed and was generally consistent with the change in the malondialdehyde content. Both increased markedly under drought stress, and the membrane permeability and malondialdehyde content under moderate drought were significantly higher than those in the control, but markedly lower than those under severe drought. This shows that malondialdehyde accumulated in large quantities under drought stress, causing progressively more damage to the cell membrane and increased its permeability. After rehydration in the early stages of drought, leaf RWC achieved a full recovery, malondialdehyde and membrane permeability decreased markedly, and there was no remarkable difference in membrane permeability among different treatments. The results showed that rehydration reduced membrane lipid peroxidation and membrane permeability, thereby allowing the plants to recover from the stress. After rehydration in the late stages of drought, leaf RWC increased but did not

return to the original level. The membrane permeability degraded under moderate drought, but the malondialdehyde content was markedly higher than that before rehydration, which indicated that the damage caused by moderate drought remained strong on August 23, and that the damage could not be fully alleviated after rehydration. However, the membrane permeability decreased, suggesting that full recovery might eventually be possible.

Under stress, ROS are produced, such as hydrogen peroxide ($H_2O_2$), the hydroxyl radical ($OH^-$), singlet oxygen ($O^{1-}$), and the superoxide radical ($O^{2-}$) [44]. To avoid or alleviate cell damage caused by ROS, plants stimulate their antioxidant enzyme system, and these protective enzymes are closely related to plant stress resistance. SOD activity is the first defense against membrane lipid peroxidation induced by ROS, and catalyzes $O^{2-}$ dismutation and conversion into $H_2O_2$ and $O_2$. CAT and POD activities are mainly responsible for removing this $H_2O_2$, thereby further lowering ROS levels [45]. In our study, POD activity was less than 0.5 U $g^{-1}$ FW throughout the drought process and rehydration, indicating a relatively low effect of POD activity on drought resistance. On August 11, SOD activity was greater than 130 U $g^{-1}$ FW and CAT activity was greater than 80 U $g^{-1}$ FW. With prolonged drought stress, CAT activity continued to increase, whereas SOD activity first increased and then decreased, and with the highest SOD activity on August 17. These results show that SOD and CAT activities played important roles during the early stage of drought. However, in the later stages of drought, the ability of SOD activity to protect the plants against drought was weakened, but CAT activity still played an important protective role. On August 23, the CAT activity under moderate drought was lower than that in the control, probably because the severity of the drought exceeded the drought tolerance of *A. halodendron*. The SOD and CAT activities after rehydration decreased, which suggests weakening of the drought stress after rehydration.

Osmotic regulation is an important drought adaptation mechanism for plants. After plants are subjected to water stress, osmoregulatory substances are produced to make the osmotic potential more negative, thereby maintaining cell turgor and normal physiological activity, which raises the plant's drought resistance. Plants with strong drought resistance have better osmotic adjustment ability than plants with weak resistance, although the adjustment ability decreases and may even be lost under severe drought [46]. We found that the soluble protein content increased with increasing drought duration, but the content under moderate drought on August 23 was lower than in the control, indicating that the soluble proteins played an active role during the early phase of drought resistance, but that their effectiveness decreased under severe stress. Soluble proteins are a storage form of nitrogen in plants, and their accumulation under stress promotes osmotic regulation and rapid recovery of growth after the stress is relieved. The change of the soluble protein content under drought stress is often accompanied by changes in the protein component [12]. Whether the increased soluble proteins content in *A. halodendron* that we observed is related to the synthesis of new osmoregulatory proteins in response to the stress needs further study. After drought stress, the soluble sugar content of *A. halodendron* increased, indicating that these sugars play a role in tolerance of drought stress. The soluble sugar content under moderate drought on August 23 was significantly higher than that in the control, but was less than that on August 17, which suggests that the plants were unable to continue producing enough of these sugars to fully protect themselves against the drought stress. The free proline content of *A. halodendron* increased markedly with increasing duration and intensity drought stress. Free proline played an important role important role in its defense mechanism, given the strong ability of this species to resist drought stress. Drought stress decreased after rehydration and levels of all three osmotic regulators were all lower than they were before rehydration. This suggests that the osmoregulatory substances promoted osmotic adjustment under drought stress, and that this adjustment protected normal physiological processes during the drought stress.

## 5. Conclusions

Our results demonstrate that *A. halodendron* has a strong ability to endure drought stress. *A. halodendron* protected itself against ROS by increasing POD and CAT activities, and reduced cell

membrane damage to survive drought stress. At the same time, it maintained normal cell processes by increasing the contents of soluble proteins, soluble sugars, and free proline. *A. halodendron* was able to tolerate a longer period under moderate drought, as all variables that we measured showed signs of recovery towards pre-drought levels after rehydration. Our results also showed that the three osmoregulatory substances were positively correlated with POD activity, but were negatively correlated with SOD activity. Peroxidation of membrane lipids increased with increasing damage to the cell membranes as the drought duration and severity increased. Severe drought caused its leaves to wilt on 23 August 2018, and the plants failed to recover even after rehydration. The stress resistance mechanism of *A. halodendron* in future research will develop from the physiological and ecological level to the molecular level.

**Author Contributions:** Conceptualization, X.Z. and Y.L. (Yuqiang Li); Investigation, J.C. and Y.L. (Yongqing Luo); Writing—original draft preparation, J.C.; Software, Y.Z. and Z.N.; Writing—Review and editing, R.W. and J.C.; Data curation, P.W. and A.C.; All authors have read the final manuscript and approved the submission.

**Funding:** This study was financially supported by the National Basic Resources Investigation Program of China (2017FY100200), the National Natural Science Foundation of China (grants 31500369 and 41371053) and the "One Hundred Talent" Program (Y551821001) of the Chinese Academy of Sciences.

**Acknowledgments:** We thank all members of Naiman Desertification Research Station, Chinese Academy of Sciences, for their help in field and laboratory work. We also wish to thank the anonymous reviewers for their valuable comments on the manuscript.

**Conflicts of Interest:** The authors declare no conflict of interest.

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
