# Peer review of "Effects of Drought and Rehydration on the Physiological Responses of Artemisia halodendron"

_water, doi:10.3390/w11040793_

Round 1
Reviewer 1 Report
· Change the keywords that must be different from the title
· Line 56- efficiency
· Is A. halondendron is an psammophytes species?
· Line 98: Artemisia halodendron Turcz. ex Besser
· How was the water capacity determined?
· What was the final irrigation volume?
· Line 128: Methodology of POD
· Line 137: LSD is a weak statistical method.
· Line 138: How do you process the images? Do you use an imagining software? If yes please explain this process in the M&M.
· Results: Please verify the statistics in all graphics.
· Line 140-141: Rewrite the paragraph
· How was the statistics calculated with the missing data? Specify
· Line 148-149: mg g-1 FW
· Line 168-173: Why are the Fv/Fm values in the control so low? How do you explain it?
· Line 179-182: Standardize the decimals
· Line 191/199: mg g-1 FW
· Line 202: Was the higher enzymes activity in drought stress plants reflected at the protein level?
· Line 228: The increase (0.3%) compared to August 11 is not significant in SD
· Line 234: Why always write the scientific name of the species since it is the only species studied? You can omit it.
· Line 238: mg g-1 FW
· Fig 4: Check the statistics
· Line 256: mg g-1 FW
· Line 263: mg g-1 FW
· Line 266/267: µg g-1 FW
· The discussion must be reviewed in view of the new results
Author Response
Dear reviewer:
Thank you for your comments concerning our manuscript entitled “Effects of drought and rehydration on the chlorophyll fluorescence parameters and physiological responses of Artemisia halodendron”. Those comments are very helpful for revising and improving our paper, as well as the important guiding significance to our researches. We have studied comments carefully and made correction. The main corrections in the paper and the responds to the reviewers¢ comments are as follows.
Change the keywords that must be different from the title
We have re-written the keywords according to the reviewer’s comments. The keywords are Horqin sandy land; Hydration-dehydration; Chlorophyll fluorescence; Lipid peroxidation; Antioxidant enzyme activity; Solute accumulation.
Line 56- efficiency
We are very sorry for our incorrect writing and have changed “Sefficiency” to “efficiency”.
Is A. halondendron is an psammophytes species?
Artemisia halodendron is a typical sandy semi-shrub and a good sand - fixing pioneer plant.
Line 98: Artemisia halodendron Turcz. ex Besser
We have added Turcz. ex Besser after Artemisia halodendron.
How was the water capacity determined?
Its field capacity water content (FC) was 13.0% (weight basis). The soil water content prior to planting was 21.1% (weight basis), equivalent to 9.6kg oven-dry soil. The pots were covered with plastic film to restrict evaporation from the soil surface and to minimize the radiant heating of the containers.
What was the final irrigation volume?
When re-watering, the final irrigation volume was 2L, the plants were well watering, superfluous water drained through holes in the pot bottom.
Line 128: Methodology of POD
Line 137: LSD is a weak statistical method.
Considering the reviewer’s suggestion, we have added much more detailed description on methods and analyses in the manuscript.
Line 138: How do you process the images? Do you use an imagining software? If yes please explain this process in the M&M.
The image was conducted using Sigma Plot 12.5 software.
Results: Please verify the statistics in all graphics.
We have verified the statistics according to the reviewer’s comments.
Line 140-141: Rewrite the paragraph
Considering the reviewer’s suggestion, we have rewritten the paragraph.
How was the statistics calculated with the missing data? Specify
Because the plants were dead on August 23 under the severe drought treatment, thus it was not possible to measure the related indicators, so we only analysed the data on August 11, 17, 18.
Line 168-173: Why are the Fv/Fm values in the control so low? How do you explain it?
Fv/Fm reflects the primary light energy conversion efficiency, and its decrease indicates that the PSII reaction centre has been damaged. One reason for its low value is that the drought has lasted for a long time by August 23, the other reason is that Fv decreases significantly with the prolonged drought time.
Line 179-182: Standardize the decimals
We have made correction according to the reviewer’s comments.
Line 202: Was the higher enzymes activity in drought stress plants reflected at the protein level?
It is really true as reviewer suggested that the higher enzymes activity in drought stress plants reflected at the protein level. In this manuscript, the correlation between the soluble proteins and three antioxidant enzymes was different, this may be related to the species, region and time period of the study. In Luo’s study (Reference 17), the soluble proteins content was significantly positively correlated with CAT activity, but was not correlated with POD and SOD activities.
17. Luo, Y.Y.; Zhao, X.Y.; Zhou, R.L.; Zuo, X.A.; Zhang, J.; Li, Y.Q. Physiological acclimation of two psammophytes to repeated soil drought and rewatering. Acta Physiologiae Plantarum 2011, 33, 79-91.
Line 228: The increase (0.3%) compared to August 11 is not significant in SD
There was no significant difference in CAT activity under severe drought on August 11 and 17, and bars were labelled with lowercase letter a. Bars labelled with different capital letters differ significantly among drought treatments on a given day.
Line 234: Why always write the scientific name of the species since it is the only species studied? You can omit it.
As reviewer suggested that we have deleted A. halodendron in many parts of the manuscript.
Fig 4: Check the statistics
Line 148-149: mg g-1 FW
Line 191/199: mg g-1 FW
Line 238: mg g-1 FW
Line 256: mg g-1 FW
Line 263: mg g-1 FW
Line 266/267: µg g-1 FW
Considering the reviewer’s suggestion, we have checked the Fig 4. The soluble sugars and proline were expressed as μg per gram of FW and the soluble proteins was expressed as milligram per gram of FW. We referred to the following literature:
Luo, Y.Y.; Zhao, X.Y.; Zhou, R.L.; Zuo, X.A.; Zhang, J.; Li, Y.Q. Physiological acclimation of two psammophytes to repeated soil drought and rewatering. Acta Physiologiae Plantarum 2011, 33, 79-91.
The discussion must be reviewed in view of the new results
We have reviewed the discussion according to the reviewer’s comments.
We tried our best to improve the manuscript and made some changes in the manuscript. These changes will not influence the content and framework of the paper. And here we did not list the changes but marked in red in revised paper.
Once again, thank you very much for your comments and suggestions.

Reviewer 2 Report
The manuscript is interesting but in my opinion does not have huge cognitive value. Individual sections of the MS differ in quality (e.g. very good Introduction, rather poor Conclusions), please revise the paper and change accordingly. Please find more detailed remarks below:
Title:
You may want to remove the phrase „chlorophyll fluorescence parameters” , as they are part of physiological response, and the paper does not highlight photochemical activity of PSII, presenting only Fv and Fv/Fm. Furthermore final conclusions refer mainly to antioxidant enzymes and osmoregulatory substances.
Abstract:
unclear fragments, e.g. line 26: SOD activity decreased under severe drought…then in:32: ..resulted from increased antioxidant enzyme activities…
Introduction:
57: „Under natural condition......during noon drought”. Does the drought only occur at noon? It is hard to verify as the references are in Chinese.
Moreover, some fragments relating similar responses in other plants should be moved to Discussion.
Please add references after: (79): However, few studies have been reported…
Materials and methods:
Wasn’t the root system injured during transplantation of 50 cm high plants? Did the plant fit a 21 cm high pot? Was the sand that filled the vases well mixed?
The plants were divided into groups on 1 August, so watering could be limited only from that date onwards. How long did it take until the moisture content dropped down to the reported values of 40-45% and 20-25%?
Methods and analyses need much more detailed description. Please inform the readers from which plant sections the analyzed material was collected.
There is no information about plant hydration status (e.g. RWC, leaf water potential)
Explanation for MDA is lacking, the abbreviation should have been introduced earlier and it is not used after introduction.
Results:
The word “significant” is overused.
In general, if an increase/decrease is observed but is not statistically significant, it shall not be presented as an increase or decrease, e.g. line 144: “...increased slightly but not significantly”
Results of biochemical analyses in drought-related experiments should be converted into DW, as conversion into FW might be misleading, e.g. the same leaf has lower chlorophyll content when hydrated than when dry - in conversion to FW.
174: (and in other places): “change in malondialdehyde” should be “change in malondialdehyde content”
179 and other lines: 67.966% - such precision is not necessary
Charts: hardly legible, the groups CONTROL, MD, SD could be moved sideways
201: “change in antioxidant enzymes”: activity? Moreover, It would be best to convert this value to protein.
Soluble proteins: although they MAY function as additional osmoregulators, I do not think this is their main role during water stress. Osmotic adaptation can probably be more easily achieved by a change in concentrations of selected ions or amino acids.
261: difference….different, …..difference….
Correlation analysis: Please provide data in correct order (starting from chlorophyll) and without repetitions (the word proteins is lacking in the first column of the table).
298: in the early stage…: but abstract mentions increased chlorophyll content at the beginning of drought…
299: Which indicate: should be: which may indicate…
255: activate antioxidant enzyme system – this system works even in stress-free conditions, maybe stimulate would be a better term
356: Style
390: Style
Conclusions: please revise. It should contain only the most important findings that provide answers to the study hypotheses. Please identify the mechanism that is/may be responsible for A. holodendron adaptation to soil drought based on your study.
References: I understand that because of the subject matter, some references were available in Chinese but please replace at least some of general references with papers in English.
A number of linguistic and editorial errors, eg.:
incorrect abbreviations - didn't (Abstract)
357: analyzes instead of catalyzes
repetitions - sandy lands line 75 and 76
style - please find some synonyms for increase/decrease, as using the same words makes the text highly repetitive and hard to follow
grammar - line150 "chlorophyll content was lower than those...",
312-314: verb form inconsistency "...reduced the ability ... and leading to...",
348-352: too long sentences make the flow of thought incomprehensible,
Conclusions: tolerate requires a complement and is incorrectly used in Conclusions (l. 402, 405)
punctuation - line 310 and many other places: Artemisia. halodendron
I hope these remarks will prove useful and help you improve your manuscript.
Author Response
Dear reviewer:
Thank you for your comments concerning our manuscript entitled “Effects of drought and rehydration on the chlorophyll fluorescence parameters and physiological responses of Artemisia halodendron”. Those comments are very helpful for revising and improving our paper, as well as the important guiding significance to our researches. We have studied comments carefully and made correction. The main corrections in the paper and the responds to the reviewers¢ comments are as follows.
The manuscript is interesting but in my opinion does not have huge cognitive value. Individual sections of the MS differ in quality (e.g. very good Introduction, rather poor Conclusions), please revise the paper and change accordingly. Please find more detailed remarks below:
Title:
You may want to remove the phrase „chlorophyll fluorescence parameters” , as they are part of physiological response, and the paper does not highlight photochemical activity of PSII, presenting only Fv and Fv/Fm. Furthermore final conclusions refer mainly to antioxidant enzymes and osmoregulatory substances.
We have re-written the title according to the reviewer’s comments. The title is Effects of drought and rehydration on the physiological responses of Artemisia halodendron.
Abstract:
unclear fragments, e.g. line 26: SOD activity decreased under severe drought…then in:32: ..resulted from increased antioxidant enzyme activities…
We are very sorry for our incorrect writing, the statements of “resulted from increased antioxidant enzyme activities” were corrected as “resulted from increased both studied enzyme (POD and CAT) activities”.
Introduction:
57: „Under natural condition......during noon drought”. Does the drought only occur at noon? It is hard to verify as the references are in Chinese.
We are very sorry for our negligence, we have changed “noon drought” to “drought stress”.
Moreover, some fragments relating similar responses in other plants should be moved to Discussion.
As reviewer suggested that we have remove the effect of drought stress on chlorophyll contents in three chickpea cultivars to the discussion.
Please add references after: (79): However, few studies have been reported…
Considering the reviewer’s suggestion, we have selected more literatures. We regret that we have not found any English literature on this theme, so we added a Chinese literature.
23. Zhou, H.Y. Drought-resistant mechanism of two edificatos in Horqin sandy land of northeast China. Bulletin of Botanical Research 2002, 22, 51-55 (in Chinese).
Materials and methods:
Wasn’t the root system injured during transplantation of 50 cm high plants? Did the plant fit a 21 cm high pot? Was the sand that filled the vases well mixed?
The root system was injured during transplantation, but the injured plants can survive. The root length of Artemisia halodendron we selected was about 500px to fit a 21 cm high pot. The sand filled in the pots was well mixed.
The plants were divided into groups on 1 August, so watering could be limited only from that date onwards. How long did it take until the moisture content dropped down to the reported values of 40-45% and 20-25%?
Its field capacity water content (FC) was 13.0% (weight basis). The soil water content prior to planting was 21.1% (weight basis), equivalent to 9.6kg oven-dry soil. The pots were covered with plastic film to restrict evaporation from the soil surface and to minimize the radiant heating of the containers.
Methods and analyses need much more detailed description. Please inform the readers from which plant sections the analyzed material was collected.
Considering the reviewer’s suggestion, we have added much more detailed description on methods and analyses in the manuscript. The surviving leaves were chosen to detect the physiological indices.
There is no information about plant hydration status (e.g. RWC, leaf water potential)
As reviewer suggested that we have added the relative water content (RWC) in the manuscript.
Leaf RWC decreased with increasing drought duration and intensity (Fig. 2A). On August 11, leaf RWC under moderate and severe drought was lower than that of control 9.6% and 15.7%, respectively, and the differences among treatments were significant. On August 17, leaf RWC under severe drought was the lowest (54.8%). On August 18, re-watering caused remarked increase, a full recovery of leaf RWC obtained, they were all greater than 87% and significantly higher than those on August 11 and 17, but there was no significant difference among treatments. On August 24, leaf RWC increased remarkably, they were significantly higher than those on August 23, but was significantly lower than those on August 18.
Explanation for MDA is lacking, the abbreviation should have been introduced earlier and it is not used after introduction.
Considering the reviewer’s suggestion, we used malondialdehyde’s original word, its abbreviation was deleted in the manuscript. The explanation for MDA was in the discussion (L323-L330).
Results:
The word “significant” is overused.
In some parts of the manuscript, we have replaced “significant” with “notable”, “distinct” and “remarkable”, and replaced “significantly” with “markedly” and “dramatically”.
In general, if an increase/decrease is observed but is not statistically significant, it shall not be presented as an increase or decrease, e.g. line 144: “...increased slightly but not significantly”
We are very sorry for our incorrect writing, the statements were corrected as “The chlorophyll content first raised slightly from August 11 to 17, and then decreased significantly from August 17 to 23”.
Results of biochemical analyses in drought-related experiments should be converted into DW, as conversion into FW might be misleading, e.g. the same leaf has lower chlorophyll content when hydrated than when dry - in conversion to FW.
Thanks for your suggestion. However, due to FW was widely used in a large number of plant physiology studies (Reference 8, 14, 16, 17, 32), so we kept using FW in the revised manuscript.
8. Sheokand, S.; Kumari, A.; Sawhney, V. Effect of nitric oxide and putrescine on antioxidative responses under NaCl stress in chickpea plants. Physiology & Molecular Biology of Plants 2008, 14, 355-362.
14. Mafakheri, A.; Siosemardeh, A.; Bahramnejad, B.; Struik, P.C.; Sohrabi, Y. Effect of drought stress on yield, proline and chlorophyll contents in three chickpea cultivars. Australian Journal of Crop Science 2010, 4, 580-585.
16. Gong, J.R.; Zhang, L.X.; Zhao, A.F.; Bi, Y.R. Elementary studies on physiological and bio-chemical anti-drought features of Artemisia ordosica. Journal of Desert Research 2002, 22, 387-392 (in Chinese).
17. Luo, Y.Y.; Zhao, X.Y.; Zhou, R.L.; Zuo, X.A.; Zhang, J.; Li, Y.Q. Physiological acclimation of two psammophytes to repeated soil drought and rewatering. Acta Physiologiae Plantarum 2011, 33, 79-91.
32. Tan, W.; Liu, J.; Dai, T.; Jing, Q.; Cao, W.; Jiang, D. Alterations in photosynthesis and antioxidant enzyme activity in winter wheat subjected to post-anthesis water-logging. Photosynthetica 2008, 46, 21-27.
174: (and in other places): “change in malondialdehyde” should be “change in malondialdehyde content”
According to the reviewer’s suggestion, we have added it in many places.
179 and other lines: 67.966% - such precision is not necessary
We have made correction according to the reviewer’s comments to standardize the decimals.
Charts: hardly legible, the groups CONTROL, MD, SD could be moved sideways
We have redrawn all the charts and enlarged the size of the icon.
201: “change in antioxidant enzymes”: activity? Moreover, It would be best to convert this value to protein.
We are very sorry for our negligence of activity, and we have added it. Your suggestion that conversion the value to protein is valuable and helpful, we will adopt it in the future study.
Soluble proteins: although they MAY function as additional osmoregulators, I do not think this is their main role during water stress. Osmotic adaptation can probably be more easily achieved by a change in concentrations of selected ions or amino acids.
Due to soluble proteins was widely used in a large number of plant physiology studies (Reference 12, 17), so we kept using soluble proteins in the revised manuscript.
12.An, Y.Y.; Liang, Z.S.; Zhao, R.K.; Zhang, J.; Wang, X.J. Organ-dependent responses of Periploca sepium to repeated dehydration and rehydration. South African Journal of Botany 2011, 77, 446-454.
17. Luo, Y.Y.; Zhao, X.Y.; Zhou, R.L.; Zuo, X.A.; Zhang, J.; Li, Y.Q. Physiological acclimation of two psammophytes to repeated soil drought and rewatering. Acta Physiologiae Plantarum 2011, 33, 79-91.
261: difference….different, …..difference….
Line 261, the statements of “There was no significant difference among the different treatments on August 11, but the difference was significant on August 17 and 23” were corrected as “There was no significant difference among the treatments on August 11, but the difference was significant on August 17 and 23”.
Correlation analysis: Please provide data in correct order (starting from chlorophyll) and without repetitions (the word proteins is lacking in the first column of the table).
We have re-written the correlation analysis according to the reviewer’s comments. We have provided the data starting from chlorophyll and added the proteins. At the same time, we have supplemented a column of RWC.
298: in the early stage…: but abstract mentions increased chlorophyll content at the beginning of drought…
We are very sorry for our incorrect writing, the previous statements were corrected as “In the early stage of drought, we found the chlorophyll content increased slightly”
299: Which indicate: should be: which may indicate…
We have added “may” to the sentence according to the reviewer’s comments.
255: activate antioxidant enzyme system – this system works even in stress-free conditions, maybe stimulate would be a better term
We have replaced “activate” with “stimulate”
356: Style
Line 356, the statements were corrected as “plants stimulate their antioxidant enzyme system, and these protective enzymes are closely related to plant stress resistance”
390: Style
Line 390, the statements were corrected as “Free proline played an important role in its defence mechanism, given the strong ability of this species to resist drought stress.”
Conclusions: please revise. It should contain only the most important findings that provide answers to the study hypotheses. Please identify the mechanism that is/may be responsible for A. holodendron adaptation to soil drought based on your study.
Considering the reviewer’s suggestion, we have re-written the conclusion.
References: I understand that because of the subject matter, some references were available in Chinese but please replace at least some of general references with papers in English.
We have changed the many general references with papers in English.
13. Anjum, F.; Yaseen, M.; Rasool, E.; Wahid, A.; Anjum, S. Water stress in barley (Hordeum vulgare L.). II. Effect on chemical composition and chlorophyll contents. Pakistan Journal of Agricultural Sciences 2003, 40, 45-49.
14. Mafakheri, A.; Siosemardeh, A.; Bahramnejad, B.; Struik, P.C.; Sohrabi, Y. Effect of drought stress on yield, proline and chlorophyll contents in three chickpea cultivars. Australian Journal of Crop Science 2010, 4, 580-585.
15. Gong, J.R.; Zhao, A.F.; Huang, Y.M.; Zhang, X.S.; Zhang, C.L. Water relations, gas exchange, photochemical efficiency, and peroxidative stress of four plant species in the Heihe drainage basin of northern China. Photosynthetica 2006, 44, 355-364.
20. Su, Y.Z.; Zhang, T.H.; Li, Y.L.; Wang, F. Changes in soil properties after establishment of Artemisia halodendron and Caragana microphylla on shifting sand dunes in semiarid Horqin sandy land, northern China. Environmental Management 2005, 36, 272-281.
25. Pavlovič, A. Photosynthetic characterization of Australian pitcher plant Cephalotus follicularis. Photosynthetica 2011, 49, 253-258.
26. Baker, N.R. Chlorophyll fluorescence: a probe of photosynthesis in vivo. Annual Review of Plant Biology 2008, 59, 89-113.
27. Izanloo, A.; Condon, A.G.; Langridge, P.; Tester, M.; Schnurbusch, T. Different mechanisms of adaptation to cyclic water stress in two South Australian bread wheat cultivars. Journal of Experimental Botany 2008, 59, 3327-3346.
35. Krause, G.H.; Weis, E. Chlorophyll fluorescence and photosynthesis: the basics. Annual Review of Plant Physiology and Plant Molecular 1991, 42, 313-349.
36. Ommen, O.E.; Donnelly, A.; Vanhoutvin, S.; Oijen, V.M.; Manderscheid, R. Chlorophyll content of spring wheat flag leaves grown under elevated CO2 concentrations and other environmental stresses within the 'ESPACE-wheat' project. European Journal of Agronomy 1999, 10, 197-203.
37.Boutraa, T. Effects of water stress on root growth, water use efficiency, leaf area and chlorophyll content in the desert shrub Calotropis procera. Journal of International Environmental Application & Science 2010, 5, 124-132.
A number of linguistic and editorial errors, eg.:
incorrect abbreviations - didn't (Abstract)
The abbreviation “didn't” was corrected as “did not”
357: analyzes instead of catalyzes
We have corrected the word according to the reviewer’s comments.
repetitions - sandy lands line 75 and 76
Line 76, “sandy lands” was deleted.
style - please find some synonyms for increase/decrease, as using the same words makes the text highly repetitive and hard to follow
In some parts of the manuscript, we have replaced “increase” with “augment” and “raise”, and replaced “decrease” with “degrade” and “reduce”.
grammar - line150 "chlorophyll content was lower than those...",
Line 150, the statements of “chlorophyll content under moderate drought was significantly lower than those in the control” were corrected as “chlorophyll content under moderate drought was significantly lower than the control”
312-314: verb form inconsistency "...reduced the ability ... and leading to...",
We have corrected the sentence according to the reviewer’s comments. The verb form was consistency “...reduced the ability ... and led to...”.
348-352: too long sentences make the flow of thought incomprehensible,
Line 348-352, the statements were corrected as “The membrane permeability decreased under moderate drought, but the malondialdehyde content was significantly higher than that before rehydration. Which indicated that the damage caused by moderate drought remained strong on August 23, and that the damage could not be fully alleviated after rehydration. However, the membrane permeability decreased, suggesting that full recovery might eventually be possible.”
Conclusions: tolerate requires a complement and is incorrectly used in Conclusions (l. 402, 405)
We have re-written the conclusion and deleted previous incorrect description according to the reviewer’s comments.
punctuation - line 310 and many other places: Artemisia. halodendron
We have corrected the word according to the reviewer’s comments.
I hope these remarks will prove useful and help you improve your manuscript.
We tried our best to improve the manuscript and made some changes in the manuscript. These changes will not influence the content and framework of the paper. And here we did not list the changes but marked in red in revised paper.
Once again, thank you very much for your comments and suggestions.

Round 2
Reviewer 1 Report
As it stands, this paper is suitable for publication for the Journal
Author Response
Dear reviewer:
We appreciate your good comments and hard work.
Once again, thank you very much for your great efforts.
Best regards
Reviewer 2 Report
To authors
In my view, the corrections improved clarity of the manuscript and enhanced its scientific value. I accept situations where the authors preferred to stick to their method of calculation (FW vs. DW, FW vs. Protein), but suggest a different approach in the next papers.
The authors clearly explained the introductory stage of the experiment (in coverletter). However, I am still concerned about starting the drought treatment at the time of transplanting the plants. It would be probably a better idea to allow the plants an adjustment phase after transplantation, as the transplantation itself may be a stressful moment. Moreover, soil/sand moisture is more evenly distributed in a pot when we dry it up than when dry soil/sand with low moisture content is watered.
I would also value a description of Artemisia halodendron root system and an information on what percent of the root (roughly) did not fit the pot (no need to add these data to the MS).
Correct representation of RWC is as follows:
RWC = [(FW – DW)/(TW – DW)] x 100%,
Also, is TW correctly described? “TW was the turgid weight after rehydration for 5 h?” This seems too short a time for full rehydration, and the referenced literature reports “overnight rehydration”.
Even after corrections, the MATERIALS AND METHODS section is rather scant. The methods are described in uneven fashion, e.g. hardly any details are provided for chlorophyll fluorescence measurements, while extraction for biochemical analysis is very precisely presented.” Leaf samples were extracted with buffer (2% polyvinylpolypyrrolidone, 50 mM phosphate) using mortar and pestle, the extraction was centrifuged at 15000 g for 20 minutes and conserved at 4 °C”.
No wavelength is given for CAT, contrary to the other measurements (POD, SOD).
Moreover, such a short methodology undivided into paragraphs makes the reader confused as to where one method ends and another begins.
Also, the sentence “The surviving leaves were chosen to detect the physiological indices” still does not say which leaves were selected for measurements (maybe the biochemical analysis involved a mix of surviving leaves but chlorophyll fluorescence was measured in specific leaves). Were they young leaves? Or old ones? Please provide more details, e.g. the second fully developed leaf from the top.
For graph description please add which dates corresponded to drought and rehydration.
Line 263: “sig remarkable nificant”
Lines 266-278: all instances of the word “significant” were replaced with “markedly” (8x) – I did not mean that “significant” was incorrect but that repetitions should be avoided and a few synonyms could be used alternatively.
- the word “activity” is still missing at a few mentions of antioxidant enzymes
Lines 301-310 – as above
Still some minor editorial mistakes present.
Author Response
Dear reviewer:
Thank you for your comments concerning our manuscript entitled “Effects of drought and rehydration on the physiological responses of Artemisia halodendron”. Those comments are very helpful for revising and improving our paper, as well as the important guiding significance to our researches. We have studied comments carefully and made correction. The main corrections in the paper and the responds to the reviewers¢ comments are as follows.
To authors
In my view, the corrections improved clarity of the manuscript and enhanced its scientific value. I accept situations where the authors preferred to stick to their method of calculation (FW vs. DW, FW vs. Protein), but suggest a different approach in the next papers.
Thanks for your kind words about our paper. We will adopt the suggestions in the next papers.
The authors clearly explained the introductory stage of the experiment (in cover letter). However, I am still concerned about starting the drought treatment at the time of transplanting the plants. It would be probably a better idea to allow the plants an adjustment phase after transplantation, as the transplantation itself may be a stressful moment. Moreover, soil/sand moisture is more evenly distributed in a pot when we dry it up than when dry soil/sand with low moisture content is watered.
Thanks for your concern. As you suggested that the transplantation itself was a stressful moment and caused mechanical damage, the transplanted plant was given more than 60 days of adjustment phase after transplantation. About sand moisture, the pots were covered with plastic film to restrict evaporation from the soil surface and to minimize the radiant heating of the containers. When watered, different water quantity and area led distinct moisture in the same pot. Thus, soil moisture is more evenly distributed in a pot when we dry it up than when dry soil/sand with low moisture content is watered.
I would also value a description of Artemisia halodendron root system and an information on what percent of the root (roughly) did not fit the pot (no need to add these data to the MS).
Artemisia halodendron is a perennial plant and its roots are concentrated in the 5-30 cm soil layer. Its horizontal roots extend as far as more than 2 meters and young seedlings have a maximum depth of about 60 cm. Its fine roots are mainly distributed 0-500px depth. The course root longer than 20 cm was cut and the fine roots well preserved. Less than 25% root was cut, roughly.
Correct representation of RWC is as follows:
RWC = [(FW – DW)/(TW – DW)] x 100%,
We have made correction according to the reviewer’s comments.
Also, is TW correctly described? “TW was the turgid weight after rehydration for 5 h?” This seems too short a time for full rehydration, and the referenced literature reports “overnight rehydration”.
We are very sorry for our incorrect writing, the statements were corrected as “TW was the turgid weight after rehydration in a petri dish covered with nylon mesh for 12 h until the weight was constant”.
Even after corrections, the MATERIALS AND METHODS section is rather scant. The methods are described in uneven fashion, e.g. hardly any details are provided for chlorophyll fluorescence measurements, while extraction for biochemical analysis is very precisely presented.” Leaf samples were extracted with buffer (2% polyvinylpolypyrrolidone, 50 mM phosphate) using mortar and pestle, the extraction was centrifuged at 15000 g for 20 minutes and conserved at 4 °C”.
Considering the reviewer’s suggestion, we have added much more detailed description on methods for chlorophyll fluorescence measurements. Chlorophyll fluorescence parameters were determined by chlorophyll fluorescence instrument (Hansatech, England) [26]. The initial fluorescence (Fo) was determined after dark adaptation of leaves for 30 min, and then the maximum fluorescence (Fm) was determined by irradiation of saturated pulses (2800 μmol/ (m2.s)). Subtracting Fo from Fm was Fv.
No wavelength is given for CAT, contrary to the other measurements (POD, SOD).
As reviewer suggested that we have added the wavelength (240 nm) for the determination of CAT activity.
Moreover, such a short methodology undivided into paragraphs makes the reader confused as to where one method ends and another begins.
We have redevided the paragraph according to the reviewer’s comments. The former paragraph is a method for determining malondialdehyde content and antioxidant enzyme activity, and the latter paragraph is a method for determining the content of osmotic regulators.
Also, the sentence “The surviving leaves were chosen to detect the physiological indices” still does not say which leaves were selected for measurements (maybe the biochemical analysis involved a mix of surviving leaves but chlorophyll fluorescence was measured in specific leaves). Were they young leaves? Or old ones? Please provide more details, e.g. the second fully developed leaf from the top.
Considering the reviewer’s suggestion, the statements were corrected as “The surviving mature leaves (completely developed leaves from the middle part of the stem) were chosen to detect the physiological indices”.
For graph description please add which dates corresponded to drought and rehydration.
According to the reviewer’s suggestion, we have added “August 11, 17 and 23 corresponded to drought, August 18 and 24 corresponded to rehydration” in graph descriptions.
Line 263: “sig remarkable nificant”
We are very sorry for our negligence, we have changed “sig remarkable nificant” to “remarkable”.
Lines 266-278: all instances of the word “significant” were replaced with “markedly” (8x) – I did not mean that “significant” was incorrect but that repetitions should be avoided and a few synonyms could be used alternatively.
As reviewer suggested that we have changed “markedly” to “evidently”. The words “significantly”, “markedly” and “evidently” were used interactively.
- the word “activity” is still missing at a few mentions of antioxidant enzymes
Lines 301-310 – as above
Lines 301-310, considering the reviewer’s suggestion, we have added activity and activities.
Still some minor editorial mistakes present.
We have added the wavelength for the determination of malondialdehyde content in the method.
Line 56, “of” was deleted.
we have added activity and activities which were missing in the whole manuscript.
We have replaced “markedly” with “evidently” in some parts of the manuscript.
We tried our best to improve the manuscript and made some changes in the manuscript and we marked using “Track Changes” function in revised paper.
Once again, thank you very much for your comments and suggestions.
